# Discarded but Not Dismissed: A Comprehensive Study of the Feeding Habits of the Brown Comber (*Serranus hepatus*, (Linneaus 1758)) in the Gulf of Cádiz (NE Atlantic)

Sara Madera-Santana [1], Carlos Rodríguez-García [1,2], Jairo Castro-Gutiérrez [1,3],
Ángel Rafael Domínguez-Bustos [1,*] and Remedios Cabrera-Castro [1,2]

[1] Departamento de Biología, Facultad de Ciencias del Mar y Ambientales, Campus Universitario de Puerto Real, Universidad de Cádiz, 11510 Puerto Real, Spain; sara.madesan@alum.uca.es (S.M.-S.); carlos.rodriguezgarcia@uca.es (C.R.-G.); jairo.castro@uca.es (J.C.-G.); reme.cabrera@uca.es (R.C.-C.)
[2] Instituto Universitario de Investigación Marina (INMAR), Campus de Excelencia Internacional del Mar (CEIMAR), 11510 Puerto Real, Spain
[3] Departamento de Ciencias Agroforestales, Escuela Técnica Superior de Ingeniería, Universidad de Huelva, 21007 Huelva, Spain
* Correspondence: angelrafael.dominguez@gm.uca.es

**Abstract:** The brown comber (*Serranus hepatus*) is a small benthopelagic species with no commercial value, primarily caught by bottom trawls as a by-catch. In this work, we studied the feeding habits of this species. For this purpose, samples were obtained from the trawl fleet within the different editions of the ECOFISH project carried out between 2019 and 2022. A total of 1534 individuals were analyzed. In the diet analysis, various factors were considered, such as the season, the depth, and the time of day of the capture, as well as the size range of the individuals caught. For the feeding analysis, different indexes were calculated, such as the vacuity index (%Vi) and index of relative importance (%IRI). The size range of the specimens was between 3.2–16.3 cm, and the weight was between 1.02–39.73 g. Of the stomach content analyzed, 49.7% of the stomachs were found to be empty. The resources with the greatest importance in the diet of the brown comber were from the crustacean group, especially mysidaceans and decapods. There were differences in the diet according to season, depth, and size; however, there was no variation in diet by the time of day.

**Keywords:** feeding; trawl fisheries; serranidae; *Serranus hepatus*

**Key Contribution:** The average size of the brown comber in the Gulf of Cádiz was 8.99 cm, with a size range between 3.2–16.3 cm. The most consumed prey by *Serranus hepatus* were crustaceans and, within that group, mysidaceans. The diet of *Serranus hepatus* shows significant variations between seasons, fishing depths, and size ranges.

## 1. Introduction

Understanding fish feeding habits is vital as they significantly influence survival, growth, and reproduction through the energy and nutrients gained from food. Feeding studies offer insights into biomass consumed, comparisons of predator and fishing mortality, and the link between recruitment variability and predation [1]. They are also instrumental in exploring fish population dynamics [2].

The brown comber (*Serranus hepatus* (Linnaeus, 1758)) is a small benthopelagic species of the family Serranidae. Due to its small size, this species has no commercial value [3,4] unlike the two other congeneric species (*S. scriba* and *S. cabrilla*) [5]. It has a brownish-yellow color, and the distinctive feature of the species is a round black spot on the rays anterior to the dorsal fin [6]. It can be found throughout the eastern Atlantic, from Portugal to the Canary Islands and Senegal, as well as in the Mediterranean [7] and recently in the Black Sea [8]. It inhabits sandy and muddy bottoms, predominantly at depths of 30 to 100 m,

seldom exceeding 200 m, and is also known to inhabit seagrass meadows like *Posidonia oceanica* [4,9–12]. This species is characterized by synchronous hermaphroditism and its gonads are ovotestis in which the male and female portions are in the same gonad [13,14], although it is challenging to distinguish at a macroscopic level; histological techniques are necessary for its determination.

There are several studies in the Mediterranean Sea about brown comber diet and feeding habits [4,5,15,16] whereby stomach content analysis revealed that the brown comber feeds mainly on decapods and juveniles of small fishes, like species of the genus Gobiidae.

The ECOFISH project, including its different phases (ECOFISH 2, ECOFISH +, and ECOFISH 4.0), promotes sustainable fisheries in the Gulf of Cádiz. Throughout its various phases, the project has evaluated and detailed the discards generated by the trawling fleet, undertaken initiatives related to the interaction between fisheries and seabirds, and addressed the issue of marine litter gathered by the trawling fleet in the Gulf of Cádiz [17]. Among the species identified in the different phases was the brown comber; this is because in the eastern Atlantic, it is primarily caught as a by-catch species by small coastal and bottom trawls [18], but total catches and the level of exploitation of the species in the area are unknown, and information on its ecology and biology in the Atlantic zone is very scarce. To fully comprehend the role of the brown comber in the ecosystem, the extent of its exploitation, and to manage it effectively, it is imperative to delve deeper into its diet behavior. Therefore, this study carried out in the Gulf of Cádiz sets out to analyze the variations in the species' diet based on factors such as season, depth, size, and time of capture, offering a comprehensive view of their feeding habits and how they adapt to their environment.

## 2. Materials and Methods

### 2.1. Study Area and Sampling Details

The work was carried out in the Gulf of Cádiz (GoC) in south-western Spain (36°51′ N, 06°55′ W, Figure 1) by artisanal bottom trawls, as categorized by the Food and Agriculture Organization of the United Nations (FAO). Sampling was carried out by the local artisanal trawl fleet operating in this area. Every fishing trip consisted of three different haul periods: the first haul (night) from 5 a.m. to 9 a.m.; the second haul (morning) from 9 a.m. to 1 p.m.; and the third haul (afternoon) from 1 p.m. to 5 p.m. The overall depth ranges from 15 to 550 m, though fishing primarily occurs at depths of 40–150 m. The fishing trips were carried out in four depth ranges: 0 to 50 m, 50 to 100 m, 100 to 150 m, and >150 m. Samples were taken monthly under the framework of the ECOFISH, ECOFISH 2, ECOFISH +, and ECOFISH 4.0. projects, except for 15 September to 31 October due to the trawl closure months in the Gulf of Cádiz.

### 2.2. Feeding Indices

Using the stomach contents of all of the obtained individuals, the diet of the brown comber was studied. In order to gain a comprehensive understanding of the brown comber diet, it is essential to determine the minimum quantity of stomachs required for analysis [19,20]. In this research, we evaluated the sufficiency of the brown comber specimens collected by charting the cumulative count of randomly chosen stomachs against the cumulative count of prey taxa. To prevent any sampling order bias, the stomachs analyzed were randomized 500 times. The emergence of an asymptotic curve indicated that an adequate number of samples had been gathered to accurately represent the diet of this species [21,22].

Vacuity index (Vi), gravimetric percentage (%W), numerical percentage (%N), frequency of occurrence (%F), relative importance index (IRI), the average weight of prey per stomach (Wm/ST), and the average number of prey per stomach (Nm/ST) were calculated.

We identified prey and classified them into taxonomic groups at the lowest possible level [23]. Stomachs with highly digested contents were classified as 'unidentified'. Once identified, we counted the prey and weighed them with a balance accurate to ±0.01 mg.

From the empty stomachs, we calculated the vacuity index (Vi):

$$\text{Vi} = \frac{\text{n° of empty stomachs}}{\text{n° of total stomachs}} \times 100$$

The percentage of the total weight of stomach contents of all stomachs analyzed, using the wet weight (g), was calculated as:

$$\%W = \frac{\text{Wet weight of item}}{\text{total weight of stomach contents}} \times 100$$

The number of prey items found in non-empty stomachs, expressed as a percentage of the total number of prey items of a resource found in each stomach:

$$\%N = \frac{\text{n° of prey items of a resource}}{\text{n° of total prey items}} \times 100$$

The percentage of stomachs containing a given type of resource or prey is expressed as:

$$\%F = \frac{\text{n° of stomachs containing a resource}}{\text{n° of total stomachs}} \times 100$$

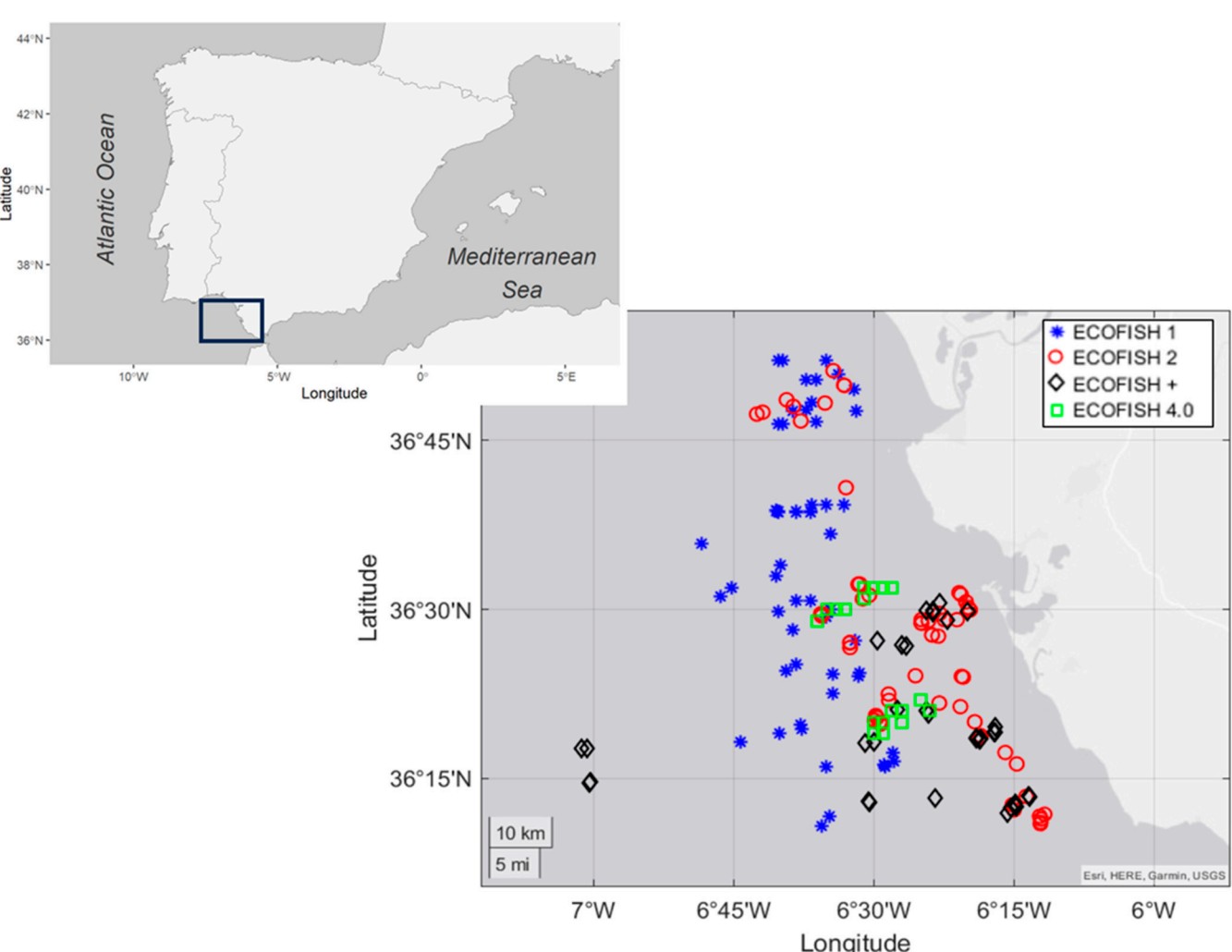

**Figure 1.** Hauls made by the trawling fleet of Sanlúcar de Barrameda and El Puerto de Santa María in the Gulf of Cádiz for the different phases of the ECOFISH project.

Finally, we calculated the importance of each prey using the relative importance index (IRI):

$$\text{IRI} = \%\text{F} \times (\%\text{N} + \%\text{W})$$

where %F is the frequency of occurrence; %N is the relative abundance; and %W is the relative weight of each prey item.

Because of synchronous hermaphroditism in the species, the specimens were not separated into males and females. To evaluate possible differences in diet with the size of the species, the specimens were divided into five size ranges. This division was based on the resulting sample size for each of the size ranges. By means of this division, 5 size ranges were obtained in which individuals of different sizes were represented with a sufficient sample size for each of the classes [16,24,25]: Range 1 [<6 cm TL), Range 2 [6, 8 cm TL), Range 3 [8, 10 cm TL), Range 4 [10, 12 cm TL), and Range 5 [12<].

### 2.3. Data Analysis

A one-way analysis of similarities (ANOSIM) was employed to assess the statistical differences in diet composition by depth, haul, season, and size. ANOSIM compares the average distances between groups with the average distances within groups. ANOSIM was performed using the Bray–Curtis distance matrix, which was generated from the prey species abundance data [26]. The Bray–Curtis distance matrix is a widely used tool in ecological studies to quantify the compositional dissimilarity between different sites or samples. This measure considers the abundance of different species, providing a more nuanced view of community composition than simple presence/absence data. The Bray–Curtis dissimilarity ranges from 0 to 1, where 0 indicates that the two samples share all species in equal proportions and 1 indicates that the two samples do not share any species. This measure is particularly suitable for diet studies, as it can effectively capture the differences in prey composition and abundance between different groups [27]. After making the Bray–Curtis distance matrix, ANOSIM was performed. There are two outputs of ANOSIM analysis: the *p*-value indicates statistical significance of the test results and the "R" statistic. The R statistic compares the mean of ranked dissimilarities between groups to the mean of ranked dissimilarities within groups. In that way, an R-value close to 1 suggests dissimilarity between groups while an R value close to 0 suggests an even distribution of high and low ranks within and between groups [28,29].

All statistical analyses were carried out using the R software (version 4.1.3) with the significance level ($\alpha$) set at 0.05.

## 3. Results

### 3.1. Fish Abundance and Fish Size Variability

A total of 1534 brown comber individuals were captured across 62 of the 90 hauls that were conducted. The mean size of the specimens was 8.99 ± 1.69 cm (TL ± standard deviation (S.D.)), with a minimum total length of 3.2 cm and a maximum of 16.3 cm, being the range 3 (8, 10 cm TL; *n* = 701), and range 4 (10, 12 cm TL; *n* = 408) was the most abundant (Figure 2). The mean weight was 13.37 ± 7.54 g (TW ± S.D.) with a weight range between 1.02–39.73 g.

### 3.2. Prey Content and Feeding Index

A total of 23 different items were identified in the 1534 stomachs studied. The cumulative prey curve did not reach the asymptotic stabilization (Figure 3, blue regression). Given that several species showed a low frequency of occurrence, a second cumulative prey curve was performed without prey with less than the 1% index of relative importance (IRI) (Figure 2, red regression). The second regression reached asymptotic stabilization with around 250 stomachs analyzed, suggesting that the number of stomachs analyzed was sufficient to describe the main diet composition of the brown comber.

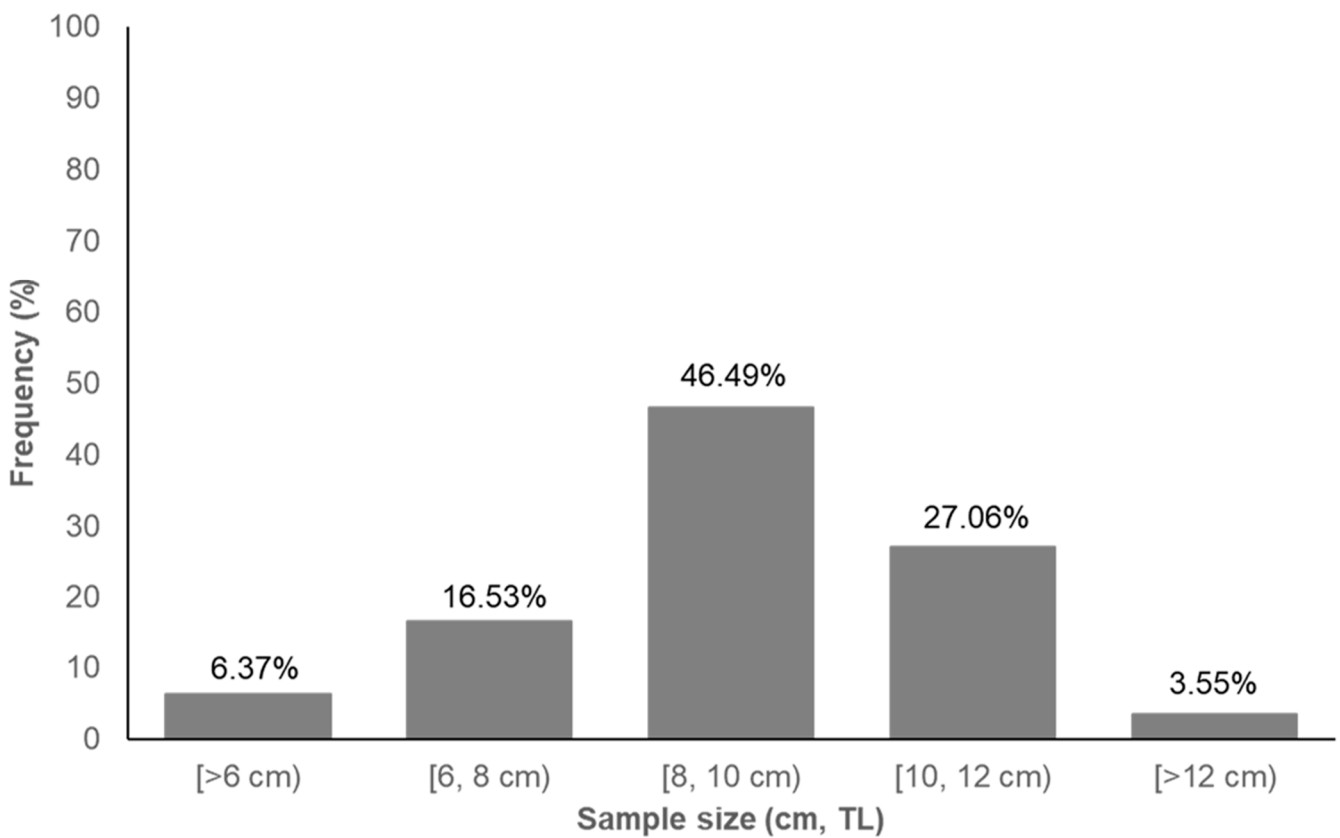

**Figure 2.** Size frequency for the brown comber (*S. hepatus*) throughout the study.

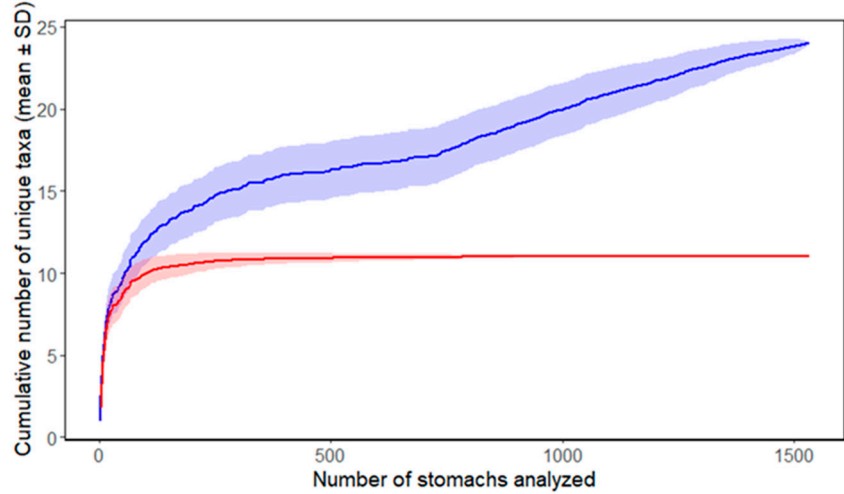

**Figure 3.** Cumulative prey curves for the total number of stomachs analyzed. The blue line represents the cumulative prey curve using all different taxa found in brown comber stomachs, and the red line represents the cumulative prey curve performed using only prey with an index of the relative importance of at least 1%.

The vacuity index showed that 49.7% of the stomachs were empty. The different items found in the stomachs are shown in Table 1, as well as the calculated index. The group of greatest relative importance in the diet of the brown comber was the crustacean group. Within this group, mysidaceans were the most consumed resource, with an IRI of 71.33%, followed by crustaceans that could not be identified in lower taxa with an IRI of 11.67%, as

well as the infraorder Caridea with an IRI of 7.70%. On the contrary, fish, molluscs, and insects presented a very small IRI (<1%).

**Table 1.** Frequency of occurrence (%F), numerical percentage (%N), gravimetric percentage (%W), relative importance index (IRI), and its percentage (%IRI) for the analyzed stomach contents of the brown comber. The different taxonomic categories identified are highlighted in bold. Unid. = unidentified.

| Prey Category | %F | %N | %W | IRI | %IRI |
|---|---|---|---|---|---|
| **Crustaceans** | | | | | |
| Mysidacea (unidentified) | 30.35 | 74.21 | 15.67 | 2728.01 | 71.33 |
| Euphasiacea (unidentified) | 0.39 | 0.81 | 0.34 | 0.45 | 0.012 |
| Amphipoda (unidentified) | 1.69 | 0.53 | 0.14 | 1.13 | 0.030 |
| Copepoda (unidentified) | 0.13 | 0.03 | 0,001 | 0.0041 | 0.0001 |
| Caridea (unidentified) | 12.84 | 3.72 | 19.23 | 294.60 | 7.70 |
| *Alpheus glaber* | 8.69 | 2.19 | 19.45 | 188.01 | 4.92 |
| *Parapenaeus longirostris* | 0.13 | 0.03 | 0.52 | 0.07 | 0.0019 |
| *Processa canaliculata* | 0.78 | 0.22 | 0.74 | 0.75 | 0.020 |
| *Plesionika heterocarpus* | 0.13 | 0.03 | 0.71 | 0.097 | 0.003 |
| Brachyura (unidentified) | 11.15 | 2.87 | 7.13 | 111.60 | 2.92 |
| *Liocarcinus* sp. | 0.26 | 0.06 | 0.34 | 0.10 | 0.0027 |
| *Goneplax rhomboides* | 4.15 | 1.25 | 8.04 | 38.55 | 1.01 |
| Galatheidae | 4.02 | 1.09 | 0.98 | 8.32 | 0.22 |
| Anomura (unidentified) | 0.13 | 0.03 | 0.05 | 0.01 | 0.0003 |
| Thalassinidea (unidentified) | 0.26 | 0.06 | 0.05 | 0.03 | 0.0008 |
| Decapoda (unidentified) | 1.30 | 0.47 | 1.13 | 2.08 | 0.054 |
| Crustacea (unidentified) | 25.42 | 6.40 | 11.16 | 446.44 | 11.67 |
| **Insects** | | | | | |
| Diptera (unidentified) | 0.13 | 0.03 | 0.001 | 0.004 | 0.0001 |
| **Molluscs** | | | | | |
| Sepiolida | 0.26 | 0.06 | 3.30 | 0.87 | 0.023 |
| Teuthida | 0.13 | 0.03 | 0.53 | 0.07 | 0.0019 |
| Cephalopoda (unidentified) | 0.26 | 0.06 | 0.64 | 0.18 | 0.0047 |
| **Teleosts** | | | | | |
| Gobiidae | 0.26 | 0.06 | 1.05 | 0.29 | 0.0075 |
| Fishes (unidentified) | 1.69 | 0.47 | 1.10 | 2.64 | 0.069 |

*3.3. Seasonal, Depth Range, Time of Day, and Size Diet Variability*

The brown comber showed a higher value of IRI for mysidaceans in spring, while in summer and autumn, decapods predominated in the diet (Figure 4A). The ANOSIM results suggested significant differences in diet composition between the seasons ($p < 0.05$).

For depth ranges between 0–50 m and 50–100 m, decapods predominated; however, for depths greater than 100 m, the relative importance of mysidaceans increased (Figure 4B). The ANOSIM results suggested significant differences in diet composition between the different fishing depths ($p < 0.05$).

In the diet by the time of day, it was observed that for morning and afternoon, the relative importance of decapods was 47.27% and 42. 99%, respectively, compared to the relative importance of mysidaceans which was 46.52% for morning and 49.87% for afternoon. As for night, the IRI for decapods was 70.58% and 19.08% for mysidaceans (Figure 4C). The ANOSIM results suggested no significant difference in diet composition between the different parts of the day ($p > 0.05$).

Finally, the analysis of the diet by size ranges shows high IRI values for mysidaceans in the first two ranges, 89.27 % IRI and 78.11 % IRI in Ranges 1 and 2, respectively. In Range 3, mysidaceans (44.76 %IRI) and decapods (46.33 %IRI) have similar importance, while in Ranges 4 and 5, it is the decapods that dominate the diet (Figure 4D). The ANOSIM results suggested significant differences in diet composition between the different fish lengths ($p < 0.05$).

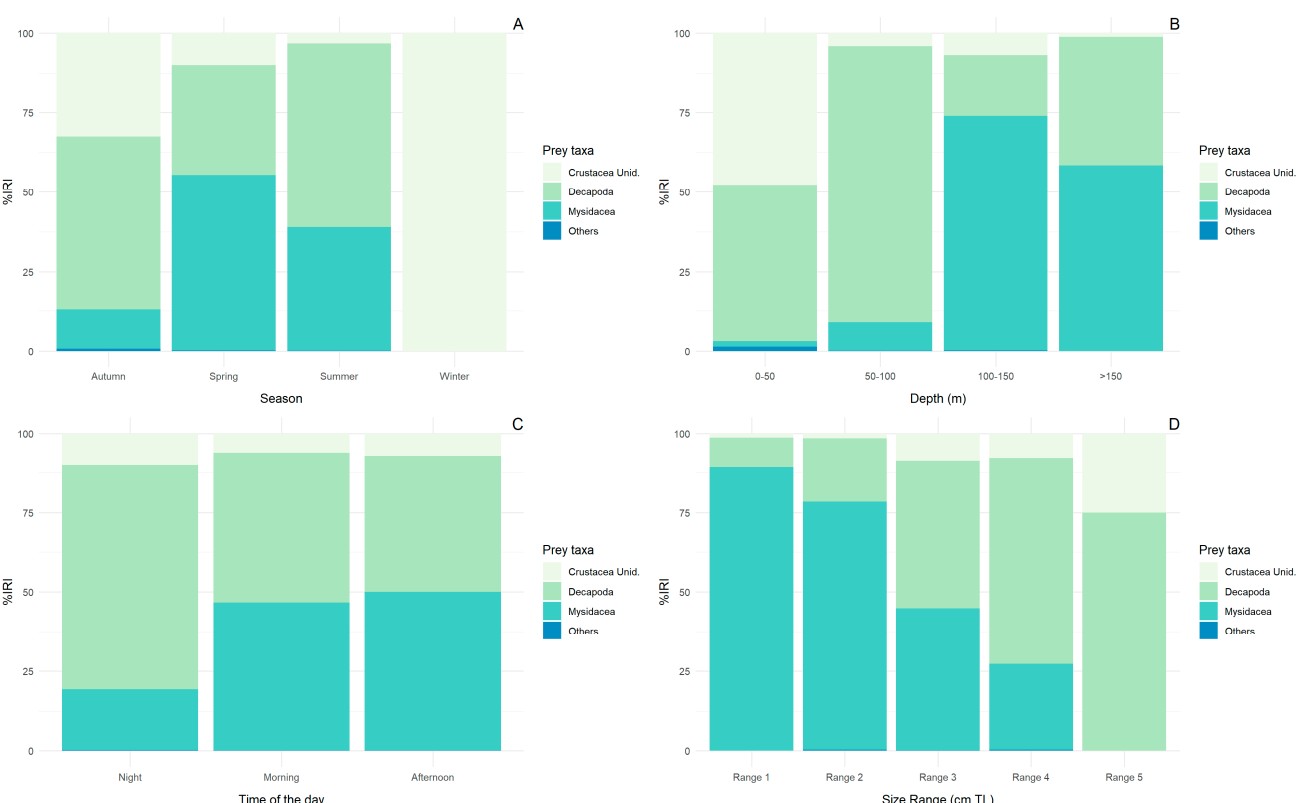

**Figure 4.** Importance of main prey groups (%IRI) by season (**A**), depth range (**B**), time of day (**C**), and size range (**D**) for the brown comber (*S. hepatus*). Unid. = unidentified.

## 4. Discussion

Our results show that the brown comber in the Gulf of Cádiz has a mean size of 8.99 cm and that its diet is mainly composed of crustaceans, with the majority being mysidaceans.

The value of the vacuity index obtained (49.7%) was lower than that obtained by Torres [18] for the same area (57%) and by Labropoulou et al. [16] in the Mediterranean (70.89%). This high percentage of empty stomachs could be because predators feeding on large prey have a distensible esophagus [30], so the expansion of gas inside the bladder, resulting from a decrease in external pressure when the trawl rises rapidly, can cause the regurgitation of food. This was observed by Labropoulou et al. [16] on the Cretan shelf, where the brown comber showed a high incidence of regurgitation, especially in larger specimens.

The identified resources belong to four major groups: Crustacea, Mollusca, Insecta, and fish. In general, crustaceans constituted 92.95% of the diet of the brown comber. These results agree with the work carried out by Torres [18] in the Gulf of Cádiz, where crustaceans were the predominant prey in the diet, as well as in the work carried out by Bilecenoglu [4] in Izmir Bay (Aegean Sea), with crustaceans making up 94.42% of the diet. However, both studies presented differences in %IRI for gobies and for the amphipod group, respectively, with both relative importance indices (9% and 33.7%) being higher than for those of this study (0.0075% and 0.03%). These differences can be explained by the number of specimens sampled: 598 by Torres [18] and 558 by Bilecenoglu [4], while in our study, 1534 individuals have been examined.

Given the scarce existing information on the brown comber in terms of diet, from this point on, the species has been compared with its congeners the painted comber *S. scriba* (Linnaeus, 1758), blacktail comber *S. atricauda* (Günther, 1874), and comber *S. cabrilla* (Linnaeus, 1758). In the present study, during summer and autumn, the diet composition was similar, with decapods predominating (%IRI = 57.85% and 54.47%, respectively), while in spring, the presence of mysidaceans increased (%IRI = 55.23%), as occurred in the study

conducted by Labropoulou et al. [16] on the Cretan shelf, in which there were no significant differences in diet throughout the year, with decapods being the most consumed prey. As for its congeners, similarities have been observed with the painted comber in Lanzarote (Canary Islands), where the seasonal analysis showed no variations in diet, with decapods being the preferred prey. However, for this species, fish were presented as the second preferred prey throughout the year [31]. In the Azores Islands [32], the opposite occurred with respect to the blacktail comber; mysidaceans were more frequent in summer with a frequency of occurrence of approximately 30%. In the other three seasons, fish dominated the diet of the species, with decapods being the secondary prey. This difference in the consumption of one or the other type of prey could point to the segregation of niche competition due to the scarcity of prey [33,34]. These variations could also be due to changes in the behavior of the species depending on the season, Carpentieri et al. [35] observed that during the summer months, the community assemblage changed between day and night, with the brown comber dominating during the day.

Diet varied with depth, with a %IRI of 49.25% for decapods and 47.71% for unidentified crustaceans being obtained. In the 50–100 m range, decapods dominated the diet with a %IRI of 86.93%, while for the 100–150 m range, mysidaceans predominated with a %IRI of 73.73%. This could be due to the vertical migration of zooplankton; when predation efficiency is low, both predator and prey are concentrated in the upper layer (nocturnal phase) and, when it is high, most of the zooplankton remain in the lower layer (diurnal phase) [36]. Therefore, at the greatest depths, mysidaceans predominated in the diet.

Diet was also studied according to the part of the day in which the specimens were captured. This was carried out to determine whether the species fed during the day or at night. It was determined that for the second haul (morning) and third haul (afternoon), the diet was composed of mysidaceans, decapods, and similarly unidentified crustaceans, with an approximate %IRI of 47%, 46%, and 6%, respectively. In the first haul (night), a higher %IRI was obtained for decapods (70.58%). This pattern in the diet according to the time of the day could be due to the vertical migrations of zooplankton, since, as previously mentioned, during the night, the mysidaceans ascend in the water column [36], thus decreasing their abundance in the benthopelagic zone, where the brown comber inhabits; thus, this could be responsible for increasing the predation of decapods. This analysis showed that the species fed during the day since the maximum values of the mean number of prey per stomach were given for the second and third hauls (1.88 and 1.77, respectively). The latter coincides with the study by Carpentieri et al. [35], in which the species showed circadian rhythms, feeding mainly from dawn to dusk, reaching their peak at midday. In contrast, except during the summer period, the brown comber showed an absence of feeding during the night. This is similar to the blacktail comber in the Azores [32] where feeding activity increased from dawn, reaching its peak at midday, and decreasing throughout the afternoon until dusk. According to Eggers [37], this is typical behavior of size-selective predators since they rely on visual cues.

Variations in diet have also been observed according to the size of the brown comber, with a substantial change from small prey in the first size ranges to larger decapods at larger sizes. These results agree with Labropoulou et al. [16], who observed a tendency to segregate trophic niches by size and age since, as the size of the fish increases, the average size of the prey also increases, thus achieving optimization of the energy expended in capturing that prey by the one that will be consumed by that prey.

The lack of data on the species in the Atlantic area makes it difficult to establish comparisons within the same area with similar environmental characteristics. However, similarities have been found with individuals inhabiting the Mediterranean in terms of feeding.

Studies on feeding habits are essential as they offer a clearer perspective on the position a species occupies within an ecosystem. Although a larger body of research is required to provide precise insights, there are indications that the diet of species like the brown comber could overlap with commercially significant species, such as hake (*Merluccius merluccius*)

juveniles [38], or other organisms with similar feeding habits. Therefore, research aimed at refining this information and discerning potential interactions between these species in the ecosystem is essential.

Studies like this one, focusing on non-commercial species that have been relatively understudied, are crucial for enhancing our understanding of their relationship with fisheries. Through analyzing discards, we gain insights into how these species interact with their environment, potentially influencing commercially valuable species and the ecosystem.

### 5. Conclusions

The brown comber, predominantly found in the Gulf of Cádiz, exhibits a diet rich in crustaceans, particularly mysidaceans. This dietary composition is influenced by depth, with a noticeable shift from decapods in shallower waters to mysidaceans in deeper regions, possibly due to the vertical migration patterns of zooplankton. Interestingly, the fish display a marked diurnal feeding pattern, peaking around midday and showing minimal nocturnal activity. This behavior suggests a reliance on visual cues for predation, a trait observed in other related species. Furthermore, as the brown comber matures, its dietary preferences evolve. Larger specimens tend to favor bigger prey, such as decapods, indicating an adaptive strategy to optimize the energy invested in hunting relative to the nutritional value of the prey. This comprehensive understanding of the brown comber's feeding habits offers valuable insights into its role within the ecosystem and potential interactions with other marine species.

**Author Contributions:** S.M.-S.: conceptualization, methodology, formal analysis, investigation, writing—original draft, writing—review and editing, and data curation. C.R.-G.: conceptualization, methodology, investigation, writing—original draft, writing—review and editing, and data curation. J.C.-G.: formal analysis, writing—original draft, and writing—review and editing. Á.R.D.-B.: investigation, writing—original draft, and writing—review and editing. R.C.-C.: investigation, writing—original draft, writing—review and editing, data curation, and supervision. All authors have read and agreed to the published version of the manuscript.

**Funding:** This research was carried out within the framework of the ECOFISH projects: eco-innovative strategies for sustainable fishing in the Gulf of Cadiz SPA. This initiative was supported by the Biodiversity Foundation, the Ministry for Ecological Transition and Demographic Challenge, through the Pleamar Program, co-financed by the European Maritime and Fisheries Fund (EMFF) [grant number: 2019-016/PV/PLEAMAR18/PT; 2020-013/PV/PLEAMAR19/PT; 2020-055/PV/PLEAMAR20/PT; 2021/PV/PLEAMAR20-21/PT; 2021-060/PV/PLEAMAR21/PT].

**Institutional Review Board Statement:** The specimens used in this work have never been subjected to animal experimentation. These specimens come from catches made by professional fishermen and are subject to European regulations on fish discards.

**Data Availability Statement:** Data will be available upon reasonable request.

**Acknowledgments:** The publication of this work was made possible thanks to the ECOFISH, ECOFISH 2, ECOFISH +, and ECOFISH 4.0. projects in collaboration with the Biodiversity Foundation of the Ministry of Ecological Transition, through the Pleamar Program, co-financed by the FEMP. The authors would like to thank the observers (Félix, Andrea, Tania, and Zaida) for going on the fishing vessels and for collecting the specimens sampled and the professional fleet of Puerto de Santa María and the Cofradía de Pescadores de Sanlúcar de Barrameda. We would also like to express our gratitude to INMAR (Instituto Universitario de Investigación Marina) for allowing us to use their laboratories to carry out all of the precision measurements.

**Conflicts of Interest:** The authors declare that they have no known competing financial interest or personal relationships that could have appeared to have influenced the work reported in this paper.

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
