# Peer review of "Discarded but Not Dismissed: A Comprehensive Study of the Feeding Habits of the Brown Comber (Serranus hepatus, (Linneaus 1758)) in the Gulf of Cádiz (NE Atlantic)"

_fishes, doi:10.3390/fishes8110541_

Round 1
Reviewer 1 Report
Comments and Suggestions for Authors
Dear Authors,
I believe that this study will make significant contributions to the understanding of the biology of this economically non-viable species in the specific context of the limited amount of nutrition studies conducted on this species in the Northeast Atlantic and its importance for the sustainable management of fisheries. This study is acceptable for publication after the major revisions mentioned in the attachment and below.
In addition, the following points should be considered:
1. Firstly, the author/authors mentioned in the abstract that the species is caught bycatch, However, there is no further discussion of the details regarding bycatch in the introduction or discussion section, such as the annual catch volume, the species caught together, regional catch quantities, etc. Therefore, I believe it is not appropriate to directly introduce the study as focusing on bycatch in the abstract. However, the authors should address and evaluate this issue in the discussion section.
2. Another point is that the exact number of stomachs analyzed to reveal the stomach contents is not clearly stated. It is necessary to provide more explanatory details in the materials and methods section regarding the representation of all species, such as how many stomachs were examined. The number of individuals used reduces the margin of error and is important.
3. Additionally, size-frequency graphs for S. hepatus should be included in the results, indicating the number of individuals identified in each size group. Presenting this information in a graphical format is necessary for a clearer understanding of the data.
4. Why weren't the individuals separated into males and females? Are there any differences in their feeding habits? It is necessary to provide this information considering the differences in feeding between males and females and their maturity stages.
5. How were the size groups (Range 1 [<6 cm TL), Range 2 [6, 8 cm TL), Range 3 [8, 10 cm TL), Range 4 [10, 12 cm TL), and Range 5 [12<]. for S. hepatus determined? This should be explained in the materials and methods section. The study did not specify the maturity stages, and it would be more accurate to separate the size groups based on maturity.
Table 1 requires significant corrections and needs to be revised again.

Author Response
Thank you to the reviewer for the constructive suggestions. We have addressed each point and outlined our responses in the attached file.

Reviewer 2 Report
Comments and Suggestions for Authors
The manuscript presents a well-done study of diet of one non-commercial fish from the Gulf of Cadiz. The volume of material is large, the methods are well described, the results are objective and the discussion is quite informative. The MS can be accepted for publication after minor changes. My comments are included in the notes to the text. These are suggestions to make the structure of the text better for its understanding.

Author Response

(The authors gave the same response as above.)

Reviewer 3 Report
Comments and Suggestions for Authors
MS: Fishes-2675327
Madera-Santana et al
Discarded but not Dismissed: A comprehensive study of the feeding
habits of the Brown Comber (Serranus hepatus, Linneaus 1758) in the Gulf of Cádiz (NE Atlantic).
Referee’s Report
I thought this ms turned up some very interesting results on a little-studied species.
However it does need some work particularly with the Abstract, which says a lot about what was done (e.g. the various indices) without saying very much at all about what was actually found (e.g. the various indices) and what this means in terms of the species’ biology and its place in the system.
While the English is generally good, there are occasional confusions (see e.g. comments lines 46, 88 ).
Some more particular comments are listed below.
Line |
Comment |
16 |
The individual ECOFISH projects do not need listed here. Leave in main text. |
17-19 |
Confusing: re-word |
21 |
Digestives? Stomachs? |
23-4 |
See comments about multivariate analysis |
34-9 |
Shorten and simplify |
46-7 |
Re-write: Posdonia is not found to depths of 100m |
57-9 |
This sentence largely repeats the first paragraph. It would be helpful here to have some indication of what is new and original about this study (other than simple geographical location), especially given that there are already several Med studies. Is the ecosystem (line 57 – this ecosystem?) significantly different? |
71 |
Line 47 seems to suggest limit is 100m: if significant numbers were found deeper (>150m), then this is an important finding (see comment above) |
88 |
Prey |
171-2 |
This is surprising in view of the very large differences (e.g. around 45% (Day) compared to over 70% (night)). Can this be explained/expanded on? |
223-44 |
Depth/Day obviously interlinked via the plankton vertical migration. Can these be untangled with ANOSIM? Or do the authors need another routine to precise this relationship? |
264 |
This is weak: can some more be said about the system and its function (e.g. inter- and antra-specific competition/predation)? |
|
|
I can recommend this ms for publication, but it would benefit from attention to the above points.

Comments on the Quality of English LanguageWhile the English is generally good, there are occasional confusion.
Author Response

(The authors gave the same response as above.)

Round 2
Reviewer 1 Report
Comments and Suggestions for Authors
Dear Authors,
"The authors have made the necessary and important corrections in the text."
Also some more minor corrections,
1. In keywords; Both 'diet' and 'feeding' should not be included;
let's remove either 'diet' or 'feeding." I think feeding more suitable.
2. In page 4, "Results" should be in next page.
3. In page 5. Figure 2, in y-exxen "It would be better to use 'Frequency (%)' instead of 'Sample size" Also should be take percentage of Frequency of sample numbers.
"I believe it is suitable for publication after the necessary corrections have been made."
Best Regards.
Author Response
We extend our gratitude to the reviewer for their valuable suggestions aimed at enhancing our article. Each comment has been addressed in the attached Word document response note.
